**Data Availability Statement:** All relevant data are within the manuscript.

**Funding:** This study was supported financially by TOYOBO Co., Ltd. PSS provided the MagDEA Dx

# The evaluation of the utility of the GENECUBE HQ SARS-CoV-2 for anterior nasal samples and saliva samples with a new rapid examination protocol

**Asami Naito[1], Yoshihiko Kiyasu[2,3], Yusaku Akashi[2,4], Akio Sugiyama[5], Masashi Michibuchi[5], Yuto Takeuchi[2,3], Shigeyuki Notake[6], Koji Nakamura[6], Hiroichi Ishikawa[7], Hiromichi Suzuki[2,3,8]***

**1** Tsukuba i-Laboratory LLP, Tsukuba, Ibaraki, Japan, **2** Division of Infectious Diseases, Department of Medicine, Tsukuba Medical Center Hospital, Tsukuba, Ibaraki, Japan, **3** Department of Infectious Diseases, University of Tsukuba Hospital, Tsukuba, Ibaraki, Japan, **4** Akashi Internal Medicine Clinic, Kashiwara, Osaka, Japan, **5** Diagnostic System Department, TOYOBO Co., Ltd., Kita-ku, Osaka, Japan, **6** Department of Clinical Laboratory, Tsukuba Medical Center Hospital, Tsukuba, Ibaraki, Japan, **7** Department of Respiratory Medicine, Tsukuba Medical Center Hospital, Tsukuba, Ibaraki, Japan, **8** Department of Infectious Diseases, Faculty of Medicine, University of Tsukuba, Tsukuba, Ibaraki, Japan

* hsuzuki@md.tsukuba.ac.jp

## Abstract

### Introduction

GENECUBE® is a rapid molecular identification system, and previous studies demonstrated that GENECUBE® HQ SARS-CoV-2 showed excellent analytical performance for the detection of severe acute respiratory syndrome coronavirus-2 (SARS-CoV-2) with nasopharyngeal samples. However, other respiratory samples have not been evaluated.

### Methods

This prospective comparison between GENECUBE® HQ SARS-CoV-2 and reference real-time reverse transcriptase polymerase chain reaction (RT-PCR) was performed for the detection of SARS-CoV-2 using anterior nasal samples and saliva samples. Additionally, we evaluated a new rapid examination protocol using GENECUBE® HQ SARS-CoV-2 for the detection of SARS-CoV-2 with saliva samples. For the rapid protocol, in the preparation of saliva samples, purification and extraction processes were adjusted, and the total process time was shortened to approximately 35 minutes.

### Results

For 359 anterior nasal samples, the total-, positive-, and negative concordance of the two assays was 99.7% (358/359), 98.1% (51/52), and 100% (307/307), respectively. For saliva samples, the total-, positive-, and negative concordance of the two assays was 99.6% (239/240), 100% (56/56), and 99.5% (183/184), respectively. With the new protocol, total-, positive-, and negative concordance of the two assays was 98.8% (237/240), 100% (56/56), and

SV and magLEAD Consumable Kit for the evaluation of the rapid protocol with magLEAD. TOYOBO Co., Ltd., provided support in the form of salaries for authors Akio Sugiyama and Masashi Michibuchi, lecture fees for author Hiromichi Suzuki, and advisory fees for author Hiromichi Suzuki. Hiromichi Suzuki also received advisory fees from PSS. The funder did not play any additional role in the study design, data collection or analysis, decision to publish, or preparation of the manuscript.

**Competing interests:** Akio Sugiyama and Masashi Michibuchi are employees of TOYOBO CO., LTD. Hiromichi Suzuki received a lecture fee and advisory fee from TOYOBO CO., LTD. Hiromichi Suzuki also received advisory fees from PSS. No additional external funding was received for this study. This does not alter our adherence to PLoS ONE policies on sharing data and materials.

98.4% (181/184), respectively. In all discordance cases, SARS-CoV-2 was detected by additional molecular examinations.

## Conclusion

GENECUBE® HQ SARS-CoV-2 provided high analytical performance for the detection of SARS-CoV-2 in anterior nasal samples and saliva samples.

## Introduction

For the diagnosis of coronavirus disease 2019 (COVID-19), accurate and rapid laboratory testing is essential. Molecular examination using real-time reverse transcriptase polymerase chain reaction (RT-PCR) has been considered the gold standard for the identification of SARS-CoV-2 [1], and nasopharyngeal samples have been commonly used for the sample examination, which requires high-level personal protective equipment [2]. For COVID-19 testing, anterior nasal samples and saliva samples have been proposed as alternative samples [3], which can be easily obtained from patients.

GENECUBE® (TOYOBO Co., Ltd., Osaka, Japan) is a Qprobe-PCR-based automated rapid molecular identification system that can detect target genes in a short time and simultaneously analyze up to 12 samples and 4 assays in a single examination [4–9]. The system automatically performs molecular examination directly, including preparation of the reaction mixtures, and amplification and detection of target genes, in 30 minutes.

GENECUBE® HQ SARS-CoV-2 (TOYOBO Co., Ltd.) is the GENECUBE® reagent for detecting the SARS-CoV-2 gene in clinical samples. This reagent was currently approved by the Ministry of Health, Labour and Welfare in Japan in October 2020. We previously evaluated the performance of the assay using 1065 nasopharyngeal samples [10]. Compared with the reference RT-PCR assay, the overall positive- and negative concordance rates were 99.7% (95% confidence interval [CI]: 99.2%–99.9%), 100.0% (95% CI: 93.4%–100.0%) and 99.7% (95% CI: 99.1%–99.9%), respectively. All discordant samples were GENECUBE® HQ SARS-CoV-2-positive and reference RT-PCR-negative, and SARS-CoV-2 was detected by another molecular assay [10]. During the previous evaluation, samples other than nasopharyngeal samples were not used.

In the present study, we evaluated the diagnostic performance of the GENECUBE® HQ SARS-CoV-2 using anterior nasal samples and saliva samples. Additionally, we evaluated a new rapid examination protocol using GENECUBE® HQ SARS-CoV-2 for the detection of SARS-CoV-2.

## Materials and methods

The current study was performed at a drive-through PCR center in Tsukuba Medical Center Hospital (TMCH) in Tsukuba, Ibaraki Prefecture, Japan, which intensively performed sample collecting and PCR analysis with nasopharyngeal samples in the Tsukuba district [10, 11]. Patients with and without symptoms were referred from nearby clinics and a local public health center. All of the asymptomatic patients had known contact histories with COVID-19 confirmed/suspected patients.

Anterior nasal samples were prospectively collected from COVID-19-suspected or COVID-19-confirmed patients in addition to nasopharyngeal samples between 11 May 2021 and 5 July 2021, as previously performed [12]. Saliva samples were also prospectively collected

in addition to nasopharyngeal samples from COVID-19-confirmed patients between 21 April 2021 and 13 May 2021. All of anterior nasal samples and saliva samples from COVID-19-confirmed patients were obtained on the same day of nasopharyngeal sample collection.

Anterior nasal samples and saliva samples were simultaneously examined using GENECUBE® HQ SARS-CoV-2 (GENECUBE examination) and reference RT-PCR, and the concordance of the two methods was evaluated.

Informed consent was verbally obtained from patients for their participation in the respective part of the current research, and participant consent was documented in the electronic chart of each participant. Written informed consent was not obtained in order to avoid infection transmission through the consent forms. The ethics committee of Tsukuba Medical Center Hospital approved the present study (approval number: 2020–066) with the current protocol, including the method of obtaining informed consent.

This study was performed in line with the principles of the Declaration of Helsinki and adheres to the STARD reporting guidelines.

For negative saliva samples, residual frozen saliva samples collected during SARS-CoV-2 active screening at hospitalization in Tsukuba Medical Center Hospital were used for the current research after anonymization.

## Sample collection

For anterior nasal samples, a nasopharyngeal-type flocked swab (Copan Italia SpA, Brescia, Italy) was inserted to a 2 cm depth in one nasal cavity, rotated five times, and held in place for 5 seconds. The swab samples were then diluted in 3 mL of UTM™ (Copan Italia SpA) immediately after sampling, and the UTM™ was then transferred to a microbiology laboratory located next to the drive-through sampling facility of the PCR center.

After arrival, purification, and ribonucleic acid (RNA) extraction were performed with magLEAD (Precision System Science Co., Ltd., Chiba, Japan) with 200 μL of fresh anterior nasal samples. RNA was eluted in 100 μL, which was used for the GENECUBE® examination and the reference RT-PCR examination. All saliva samples were stored at −80˚C and were purified with magLEAD after preparation (Fig 1). All of the GENECUBE® examinations and reference RT-PCR examinations were performed simultaneously on the same day.

## GENECUBE® examination with magLEAD and discrepancy analysis

The sample used for the GENECUBE® examination analysis of SARS-CoV-2 was also used for the RT-PCR assays. All assays were performed with the previously described magLEAD conditions [10] (Fig 1). In addition to the standard protocol with magLEAD purification, a rapid protocol created by Hiromichi Suzuki, TOYOBO Co., Ltd. and Precision System Science Co., Ltd. were evaluated for saliva samples and samples for limit of detection (LOD) analysis. For the rapid protocol, in the preparation of saliva samples, purification and extraction processes were adjusted, and the total process time was shortened to approximately 10 minutes. The comparison of the standard protocol and the rapid protocol for each magLEAD purification processes is summarized in Table 1. The rapid protocol is commercially available in Japan as MagDEA Dx SV 200 for GENECUBE®.

If discordance was recognized between GENECUBE® and the reference RT-PCR, an additional evaluation was performed with Xpert® Xpress SARS-CoV-2 and GeneXpert® (Cepheid Inc., Sunnyvale, CA, USA) [15] analyses for anterior nasal samples according to the manufacturer's instructions documented in the package insert, and with an RT-PCR with LightMix® Modular SARS-CoV (COVID19) E-gene (Roche Diagnostics KK) [16] for saliva samples along with re-evaluation with the NIID RT-PCR method.

**(a) Standard method with magLEAD extraction for GENECUBE®**

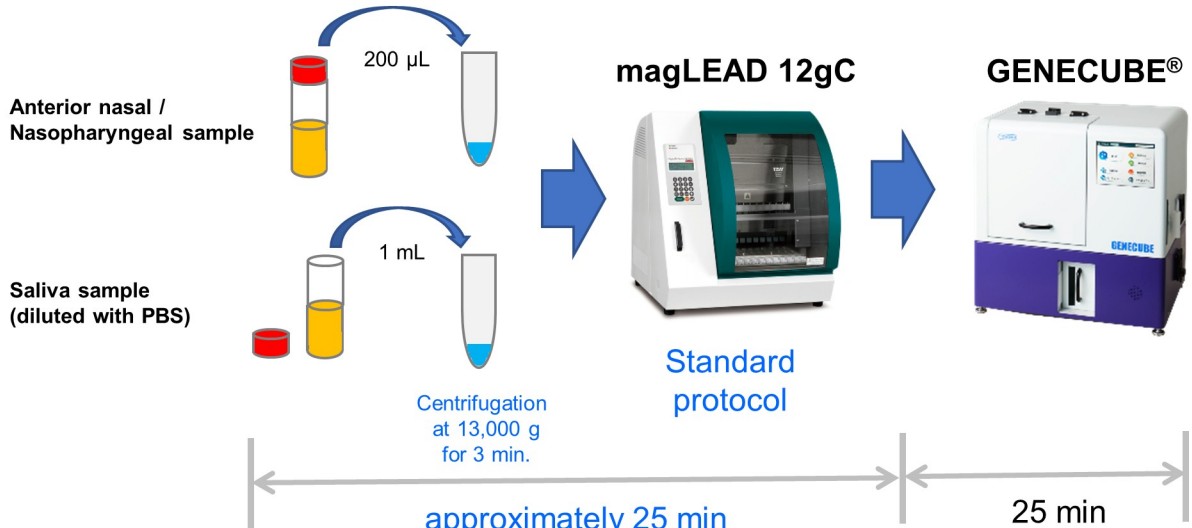

**(b) Rapid method with magLEAD extraction for GENECUBE®**

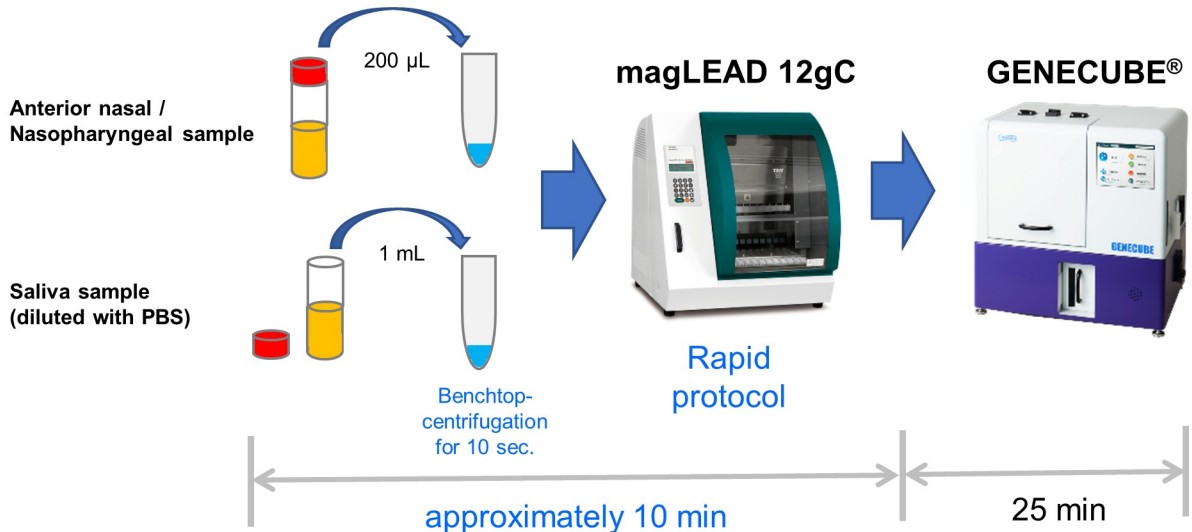

**Fig 1. Workflow of the two extraction methods for the GENECUBE® assay in this study.** The rapid method with magLEAD extraction (b) newly developed in this study takes as little as 10 min for viral RNA extraction, while the standard method (a) takes approximately 25 min. For the rapid protocol, in the preparation of saliva samples, purification and extraction processes were adjusted, and the total process time was shortened. *PBS* Phosphate-buffered saline. For magLEAD 12gC, the picture was reprinted from [13] under a CC BY license with the permission of Precision System Science Co., Ltd., 2016. For GENECUBE, the picture was reprinted from [14] under a CC BY license with the permission of TOYOBO Co., Ltd., 2021.

### Reference real-time RT-PCR method

Reference RT-PCR examinations were performed with purified samples using a method developed by the National Institute of Infectious Diseases (NIID), Japan, for SARS-CoV-2 [17, 18], which was used Briefly, 5 μL of the extracted RNA was used for one-step quantitative RT-PCR with the THUNDERBIRD® Probe One-step qRT-PCR kit (TOYOBO Co., Ltd.) and the Light-Cycler® 96 Real-time PCR System (Roche Diagnostics KK, Basel, Switzerland). A duplicate

**Table 1. The comparison of the standard and rapid protocols for each magLEAD purification process.**

| Process | Rapid protocol MagDEA Dx SV 200 for GENECUBE® | Standard protocol MagDEA Dx SV 200 |
|---|---|---|
| Lysis process | 1.5 min | 4.0 min |
| Binding to Magnetic Beads process | 1.0 min | 2.0 min |
| Washing process | 0.5 min × 1 | 1.5 min × 2 |
| Elution process | 1.0 min | 5.0 min |
| Other process: Motor operation, sample and buffer preparation, liquid dispensing, B/F separation, eluate collection, etc. | | |
| Total time | Approximately 10 min | Approximately 25 min |

For the Rapid protocol MagDEA Dx SV 200 for GENECUBE®, operation speeds, including the interval of each process, were made as fast as possible.

analysis for N2 genes was performed for the evaluation of SARS-CoV-2. EDX SARS-CoV-2 Standard (Bio-Rad Laboratories, Inc., Hercules, CA, USA) and sterile purified water (Merck & Co., Inc., Kenilworth, NJ, USA) were used as positive and negative controls, respectively. The calibration curves were generated with 5, 50 and 500 copies/reaction of EDX SARS-CoV-2 Standard.

## Estimation of the limit of detection (LOD) for GENECUBE® HQ SARS-CoV-2 with nasopharyngeal samples and saliva samples

To estimate the LOD for GENECUBE® HQ SARS-CoV-2, we made four different concentrations of samples (2500 copies/mL, 1000 copies/mL, 500 copies/mL, 250 copies/mL) with SARS-CoV-2 reference material (AccuPlex™ SARS-CoV-2 Reference Material Kit, SeraCare; SeraCare Life Sciences, Inc., Milford, MA, USA) and matrix (UTM™; three pooled nasopharyngeal samples and two pooled saliva samples). In total, six samples were made at each concentration. The GENECUBE® examination was performed four times, and the reference RT-PCR was performed twice for each sample.

## Statistical analyses

The positive-, negative-, and total concordance rates of the GENECUBE® examinations compared with the reference RT-PCR were calculated using the Clopper and Pearson methods with 95% confidence intervals. All calculations were conducted using the R 3.3.1 software program (The R Foundation, Vienna, Austria).

## Results

### Evaluation of LOD for the reference RT-PCR and GENECUBE® with SARS-CoV-2 reference material and pooled negative samples

The details of the results of the LOD evaluation for the three SARS-CoV-2 detection methods with SARS-CoV-2 reference material and pooled negative samples are listed in Table 2, summarized in Table 3 and S1 Fig.

The reference NIID real-time RT-PCR method showed positive results for all UTM-based samples (range: 250–5000 copies/mL), while the detection rate was 100% down to 1000 copies/mL for pooled nasopharyngeal samples and down to 2500 copies/mL for pooled saliva-based samples. None of the 500 copies/mL of pooled saliva-based samples and 250 copies/mL of pooled saliva-based samples were detected by the NIID real-time RT-PCR method.

**Table 2. Detailed results of the estimated LOD test for three SARS-CoV-2 detection methods.**

| Ratio of reference material and sample | Copies/mL | Sample | GENECUBE® (Standard method with magLEAD) | GENECUBE® (Rapid method with magLEAD) | Real-time RT-PCR (N2 NIID method) | |
|---|---|---|---|---|---|---|
| | | | N of detection/N of test (detection rate) | | | Ct value (Copies/test) |
| reference material: sample = 1:1 | 2500 | Total | 24/24 (100) | 24/24 (100) | 12/12 (100) | - |
| | | UTM | 4/4 (100) | 4/4 (100) | 2/2 (100) | 31.9 (48)/32.4 (34) |
| | | Pooled nasopharyngeal sample 1 | 4/4 (100) | 4/4 (100) | 2/2 (100) | 32.2 (39)/31.9 (50) |
| | | Pooled nasopharyngeal sample 2 | 4/4 (100) | 4/4 (100) | 2/2 (100) | 32.7 (28)/32.3 (36) |
| | | Pooled nasopharyngeal sample 3 | 4/4 (100) | 4/4 (100) | 2/2 (100) | 32.2 (40)/32.4 (35) |
| | | Pooled saliva sample 1 | 4/4 (100) | 4/4 (100) | 2/2 (100) | 32.3 (38)/32.5 (33) |
| | | Pooled saliva sample 2 | 4/4 (100) | 4/4 (100) | 2/2 (100) | 32.3 (37)/32.7 (28) |
| reference material: sample = 1:4 | 1000 | Total | 24/24 (100) | 24/24 (100) | 10/12 (83) | - |
| | | UTM | 4/4 (100) | 4/4 (100) | 2/2 (100) | 34.1 (11)/33.5 (16) |
| | | Pooled nasopharyngeal sample 1 | 4/4 (100) | 4/4 (100) | 2/2 (100) | 33.7 (15)/33.7 (15) |
| | | Pooled nasopharyngeal sample 2 | 4/4 (100) | 4/4 (100) | 2/2 (100) | 33.5 (16)/33.6 (15) |
| | | Pooled nasopharyngeal sample 3 | 4/4 (100) | 4/4 (100) | 2/2 (100) | 34.1 (11)/33.1 (21) |
| | | Pooled saliva sample 1 | 4/4 (100) | 4/4 (100) | 1/2 (50) | 33.3 (19)/ND |
| | | Pooled saliva sample 2 | 4/4 (100) | 4/4 (100) | 1/2 (50) | 33.8 (14)/ND |
| reference material: sample = 1:9 | 500 | Total | 23/24 (96) | 24/24 (100) | 4/12 (33) | - |
| | | UTM | 4/4 (100) | 4/4 (100) | 2/2 (100) | 34.5 (8)/35.3 (5) |
| | | Pooled nasopharyngeal sample 1 | 4/4 (100) | 4/4 (100) | 0/2 (0) | ND/ND |
| | | Pooled nasopharyngeal sample 2 | 3/4 (75) | 4/4 (100) | 2/2 (100) | 34.3 (10)/34.2 (10) |
| | | Pooled nasopharyngeal sample 3 | 4/4 (100) | 4/4 (100) | 0/2 (0) | ND/ND |
| | | Pooled saliva sample 1 | 4/4 (100) | 4/4 (100) | 0/2 (0) | ND/ND |
| | | Pooled saliva sample 2 | 4/4 (100) | 4/4 (100) | 0/2 (0) | ND/ND |
| reference material: sample = 1:19 | 250 | Total | 19/24 (79) | 18/24 (75) | 3/12 (25) | - |
| | | UTM | 4/4 (100) | 4/4 (100) | 2/2 (100) | 36.2 (3)/34.8 (7) |
| | | Pooled nasopharyngeal sample 1 | 4/4 (100) | 4/4 (100) | 0/2 (0) | ND/ND |
| | | Pooled nasopharyngeal sample 2 | 4/4 (100) | 2/4 (50) | 1/2 (50) | 35.1 (5)/ND |
| | | Pooled nasopharyngeal sample 3 | 2/4 (50) | 3/4 (75) | 0/2 (0) | ND/ND |
| | | Pooled saliva sample 1 | 3/4 (75) | 2/4 (50) | 0/2 (0) | ND/ND |
| | | Pooled saliva sample 2 | 2/4 (50) | 3/4 (75) | 0/2 (0) | ND/ND |

The Accuplex™ SARS-CoV-2 reference material (5000 copies/mL) was diluted with UTM or pooled samples and subjected to magLEAD extraction with the standard or rapid method. Each extract was then assayed four times by GENECUBE and twice by NIID RT-PCR.

*Ct* cycle threshold, *LOD* limit of detection, *ND* not detected, *NIID* National Institute of Infectious Diseases, *RT-PCR* reverse transcription polymerase chain reaction

**Table 3. Summary of the estimated LOD test results for three SARS-CoV-2 detection methods.**

| Sample (Copies/mL) | GENECUBE® (Standard method with magLEAD) | GENECUBE® (Rapid method with magLEAD) | Real-time RT-PCR (N2 NIID method) |
|---|---|---|---|
| | N of detection/N of test (detection rate) | N of detection/N of test (detection rate) | N of detection/N of test (detection rate) |
| 2500 | 24/24 (100) | 24/24 (100) | 12/12 (100) |
| 1000 | 24/24 (100) | 24/24 (100) | 10/12 (83) |
| 500 | 23/24 (96) | 24/24 (100) | 4/12 (33) |
| 250 | 19/24 (79) | 18/24 (75) | 3/12 (25) |

*LOD* limit of detection, *N* number, *NIID* National Institute of Infectious Diseases, *RT-PCR* reverse transcription polymerase chain reaction

The standard protocols with magLEAD and GENECUBE® showed positive results for all UTM-based samples. The detection rate was 100% down to 1000 copies/mL for pooled naso-pharyngeal-based samples and down to 500 copies/mL for pooled saliva-based samples. The detection rate of 500 copies/mL pooled nasopharyngeal-based samples was 91.7% (11/12).

For the rapid protocol with magLEAD and GENECUBE®, the method showed positive results for all UTM-based samples. The detection rate was 100% down to 500 copies/mL for pooled nasopharyngeal-based samples and pooled saliva-based samples.

## Comparison of the reference RT-PCR and GENECUBE® for the detection of SARS-CoV-2 with anterior nasal samples

In this study, we prospectively evaluated 359 fresh anterior nasal samples, including 59 samples with positive SARS-CoV-2 results for simultaneously collected nasopharyngeal samples (cycle threshold (Ct) < 20, n = 40; Ct ≥ 20 to < 30, n = 16; Ct ≥ 30, n = 3) (S1 Table). Of the 359 anterior nasal samples, 298 (83.0%) were obtained from asymptomatic patients.

The comparison of the reference RT-PCR and GENECUBE® (standard protocol) for the detection of SARS-CoV-2 with anterior nasal samples is summarized in Tables 4–6. For anterior nasal samples prospectively obtained from suspected COVID-19 patients (Table 4), the total, positive and negative concordance of the 2 assays were 100% (320/320), 100% (18/18) and 100% (302/302), respectively. With the addition of the enriched positive patients, the total, positive and negative concordance of the 2 assays were 99.7% (358/359), 98.1% (51/52) and

**Table 4. Concordance rate of the GENECUBE® HQ SARS-CoV-2 with real-time RT-PCR for anterior nasal samples obtained from suspected COVID-19 patients.**

| | | Real-time RT-PCR (N2 NIID method) (Anterior nasal samples) | |
|---|---|---|---|
| | | Positive | Negative |
| Standard method with magLEAD extraction for GENECUBE® (Anterior nasal sample) | Positive | 18 | 0 |
| | Negative | 0 | 302 |
| Positive concordance rate (%) | | 100 (81.5–100) | |
| Negative concordance rate (%) | | 100 (98.8–100) | |
| Total concordance rate (%) | | 100 (98.9–100) | |

*NIID* National Institute of Infectious Diseases, *RT-PCR* reverse transcription polymerase chain reaction

Data in parentheses are 95% confidence intervals.

**Table 5. Concordance rate of the GENECUBE® HQ SARS-CoV-2 with real-time RT-PCR for anterior nasal samples obtained from suspected or confirmed COVID-19 patients.**

| | | Real-time RT-PCR (N2 NIID method) (Anterior nasal samples) | |
|---|---|---|---|
| | | Positive | Negative |
| Standard method with magLEAD extraction for GENECUBE® (Anterior nasal sample) | Positive | 51 | 0 |
| | Negative | 1* | 307 |
| Positive concordance rate (%) | | 98.1 (89.7–100) | |
| Negative concordance rate (%) | | 100 (98.8–100) | |
| Total concordance rate (%) | | 99.7 (98.5–100) | |

*NIID* National Institute of Infectious Diseases, *RT-PCR* reverse transcription polymerase chain reaction

Data in parentheses are 95% confidence intervals.

*The discordant sample was tested by Xpert® Xpress SARS-CoV-2 and GeneXpert® and was positive.

100% (307/307), respectively. When GENECUBE® (standard protocol) with anterior nasal samples were compared with the reference RT-PCR with nasopharyngeal samples, the total, positive and negative concordance were 97.8% (351/359), 86.4% (51/59) and 100% (300/300), respectively.

## Comparison of the reference RT-PCR and GENECUBE® for the detection of SARS-CoV-2 with saliva samples

For the comparison between the reference RT-PCR and GENECUBE® for the detection of SARS-CoV-2, 60 frozen samples (symptomatic patients, 32; asymptomatic patients, 28) obtained from confirmed COVID-19 patients by nasopharyngeal samples and 180 frozen negative saliva samples were examined.

The evaluation of the standard protocol with magLEAD and GENECUBE® is described in Table 7, and the evaluation of the rapid protocol with magLEAD and GENECUBE® is described in Table 8. The result of one sample was invalid initially by both GENECUBE® examinations, and the sample required four-fold dilution with lysis buffer for the GENECUBE® examinations.

For the evaluation of the standard protocol with magLEAD and GENECUBE® (Table 7), the total-, positive-, and negative concordance of the two assays was 99.6% (239/240), 100% (56/56), and 99.5% (183/184), respectively.

**Table 6. Concordance rate between the GENECUBE® HQ SARS-CoV-2 with anterior nasal samples and real-time RT-PCR with nasopharyngeal samples, both of which were simultaneously obtained from suspected or confirmed COVID-19 patients.**

| | | Real-time RT-PCR (N2 NIID method) (Nasopharyngeal samples) | |
|---|---|---|---|
| | | Positive | Negative |
| Standard method with magLEAD extraction for GENECUBE® (Anterior nasal samples) | Positive | 51 | 0 |
| | Negative | 8 | 300 |
| Positive concordance rate (%) | | 86.4 (75.0–94.0) | |
| Negative concordance rate (%) | | 100 (98.8–100) | |
| Total concordance rate (%) | | 97.8 (95.7–99.0) | |

*NIID* National Institute of Infectious Diseases, *RT-PCR* reverse transcription polymerase chain reaction

Data in parentheses are 95% confidence intervals.

**Table 7. Concordance rate of the standard method with magLEAD extraction for GENECUBE® with real-time RT-PCR for saliva samples\*.**

| | | Real-time RT-PCR (N2 NIID method) | |
|---|---|---|---|
| | | Positive | Negative |
| Rapid method with magLEAD extraction for GENECUBE® | Positive | 56 | 1\*\* |
| | Negative | 0 | 183 |
| Positive concordance rate (%) | | 100 (93.6–100) | |
| Negative concordance rate (%) | | 99.5 (97.0–100) | |
| Total concordance rate (%) | | 99.6 (97.7–100) | |

*NIID* National Institute of Infectious Diseases, *RT-PCR* reverse transcription polymerase chain reaction.

Data in parentheses are 95% confidence intervals.

\* 60 frozen samples obtained from confirmed COVID-19 patients by nasopharyngeal samples and 180 negative saliva samples were used.

\*\*The discordant samples were tested by real-time RT-PCR with Roche LightMix Modular SARS and Wuhan CoV E-gene and all were positive (S3 Table).

For the evaluation of the rapid protocol with magLEAD and GENECUBE® (Table 8), the total-, positive-, and negative concordance of the two assays was 98.8% (237/240), 100% (56/56), and 98.4% (181/184), respectively.

## Discussion

During the analytical evaluation with 359 anterior nasal samples and 240 saliva samples, the GENECUBE® evaluation with GENECUBE® HQ SARS-CoV-2 showed high concordance rates compared with the reference RT-PCR. The estimated LoDs study indicated that the GENECUBE® evaluation conducted with GENECUBE® HQ SARS-CoV-2 maintained a high analytical performance for nasopharyngeal samples and saliva samples, with detection successful for as little as 1000 copies/mL for both types of samples.

In the current study concerning the GENECUBE examination, the rapid method with magLEAD extraction showed equivalent analytical performance to the standard method, both in the estimated LOD study and the comparative study with reference via real-time RT-PCR. Of note, two saliva samples obtained from COVID-19-confirmed patients were positive with the rapid protocol and negative with the standard protocol. This slight difference might be due to differences in the centrifugation condition (Fig 1) and number of washes performed (Table 1), which can result in the loss of virus and RNA; however, the difference was not proven in the estimated LOD study, and the superiority with respect to the analytical performance cannot be concluded based on the present findings.

**Table 8. Concordance rate of the rapid method with magLEAD extraction for GENECUBE® with real-time RT-PCR for saliva samples\*.**

| | | Real-time RT-PCR (N2 NIID method) | |
|---|---|---|---|
| | | Positive | Negative |
| Rapid method with magLEAD extraction for GENECUBE® | Positive | 56 | 3\*\* |
| | Negative | 0 | 181 |
| Positive concordance rate (%) | | 100 (93.6–100) | |
| Negative concordance rate (%) | | 98.4 (95.3–99.7) | |
| Total concordance rate (%) | | 98.8 (96.4–99.7) | |

*NIID* National Institute of Infectious Diseases, *RT-PCR* reverse transcription polymerase chain reaction

Data in parentheses are 95% confidence intervals.

\* 60 frozen samples obtained from confirmed COVID-19 patients by nasopharyngeal samples and 180 negative saliva samples were used.

\*\*The discordant samples were tested by real-time RT-PCR with Roche LightMix Modular SARS and Wuhan CoV E-gene and all were positive (S3 Table).

Saliva has been considered a good alternative for the detection of SARS-CoV-2 in COVID-19 patients [3] and has been widely used in COVID-19 practice. Among rapid molecular identification systems, GeneXpert® showed good analytical performance for the detection of SARS-CoV-2 with saliva samples [19]; however, the application of saliva to rapid molecular identification systems remains a challenge owing to saliva's viscosity and RT-PCR inhibition. In our current study, we used 60 saliva samples with positive nasopharyngeal sample results for SARS-CoV-2 (S2 Table), and the rapid protocol detected SARS-CoV-2 in most samples (98.3%; 59/60). The rapid protocol can detect SARS-CoV-2 with high performance in approximately 35 minutes with saliva samples, and the protocol is expected to have clinical utility, especially for the rapid accurate identification of SARS-CoV-2-infected patients with collecting nasopharyngeal samples is considered difficult. The current study used frozen saliva samples for validation. While the analytical performance for the detection of SARS-CoV-2 has been reported to be equal between fresh and frozen saliva samples [20], the freeze-thaw method has been known as one of the RNA/DNA extraction methods [21] which might influence the sensitivity [22]. Therefore, a further evaluation should be performed with fresh samples.

Anterior nasal samples are also a good alternative method for COVID-19 sampling [3]. The method has been reported as less painful and induced fewer coughs or sneezes compared with nasopharyngeal sampling [12]. The application of self-collected anterior nasal sampling has also been reported [23]. In the current study, the analytical performance of the GENECUBE® examination was almost identical to the reference RT-PCR. However, there were seven negative results using anterior nasal samples, which were obtained from patients with positive nasopharyngeal samples (S1 Table). The detection rate was 88.1% (52/59), which was inferior to that with saliva samples; however, saliva samples were not simultaneously collected with anterior nasal samples in the present study, so we cannot compare the sensitivity. In addition, we obtained anterior nasal samples from a single nasal cavity; however, the viral loads obtained with nasal sampling differ between nares [24], so sampling from both nasal cavities is preferable [25].

There are several limitations in this study that should be mentioned. First, reference real-time RT-PCR was used for the comparison, and a discrepant analysis was used for the validation, which can cause bias unless a composite reference standard is used as a reference [26–32]. While the current study evaluated the concordance with the reference real-time RT-PCR findings, the sensitivity and specificity of the GENECUBE® examination were not accurately confirmed. Second, the current research was performed at a PCR center in Japan. The influence of LODs of the GENECUBE® evaluation for genetic variants of SARS-CoV-2 was not evaluated in this study. Third, the current evaluated rapid protocol showed excellent performance for the detection of SARS-CoV-2; however, the sample size was insufficient to conclude that the protocol can be used in clinical practice; additional evaluation in studies with large samples is required. Fourth, the current GENECUBE® examination can analyze only 12 samples at a single run, and the amplification curve is not displayed. The system must be improved for better examination. Fourth, for the evaluation of the GENECUBE® examination using anterior nasal samples, the proportion of low viral load samples (Ct ≥ 30) was small, which could have improved thus concordance rates of the current study.

In conclusion, the GENECUBE® examination with GENECUBE® HQ SARS-CoV-2 provided high analytical performance for the detection of SARS-CoV-2 in anterior nasal samples and saliva samples.

## Supporting information

**S1 Fig. Detection rate of three SARS-CoV-2 detection methods for each concentration of samples.** The same set of samples was extracted with the standard or rapid method with magLEAD and analyzed with GENECUBE® and real-time RT-PCR (N2 NIID method) for

each sample concentration. The vertical axis shows the detection rate (%). The horizontal axis shows the comparison of each three SARS-CoV-2 detection methods, and each of the bar graph types shows the sample concentration.
(TIF)

**S1 Table. Results of SARS-Cov-2 detection for anterior nasal samples.**
(DOCX)

**S2 Table. Results of SARS-Cov-2 detection for saliva samples.**
(DOCX)

**S3 Table. Detailed data for the three cases with discordant findings among the three SARS-CoV-2 detection methods for saliva samples.**
(DOCX)

## Acknowledgments

For their thorough support of this study, we are very grateful to Mrs. Yoko Ueda, Mrs. Mio Matsumoto, Mrs. Asami Nakayama, and the staff in the Department of the Clinical Laboratory of Tsukuba Medical Center Hospital. We thank all of the medical institutions for providing their patients' clinical information.

## Author Contributions

**Conceptualization:** Hiromichi Suzuki.

**Data curation:** Asami Naito, Yusaku Akashi, Akio Sugiyama, Masashi Michibuchi, Yuto Takeuchi, Shigeyuki Notake, Koji Nakamura, Hiromichi Suzuki.

**Formal analysis:** Yusaku Akashi, Akio Sugiyama, Masashi Michibuchi, Hiromichi Suzuki.

**Funding acquisition:** Hiromichi Suzuki.

**Investigation:** Asami Naito, Akio Sugiyama, Masashi Michibuchi, Yuto Takeuchi, Shigeyuki Notake, Koji Nakamura, Hiromichi Suzuki.

**Methodology:** Akio Sugiyama, Masashi Michibuchi, Hiromichi Suzuki.

**Project administration:** Asami Naito, Hiromichi Suzuki.

**Resources:** Hiromichi Suzuki.

**Supervision:** Yoshihiko Kiyasu, Hiroichi Ishikawa, Hiromichi Suzuki.

**Validation:** Asami Naito, Akio Sugiyama, Masashi Michibuchi, Hiromichi Suzuki.

**Visualization:** Asami Naito, Hiromichi Suzuki.

**Writing – original draft:** Asami Naito, Yoshihiko Kiyasu, Yusaku Akashi, Akio Sugiyama, Hiromichi Suzuki.

**Writing – review & editing:** Asami Naito, Yoshihiko Kiyasu, Yusaku Akashi, Akio Sugiyama, Masashi Michibuchi, Yuto Takeuchi, Shigeyuki Notake, Koji Nakamura, Hiroichi Ishikawa, Hiromichi Suzuki.

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
