## [Decision Letter · Decision Letter 0]

21 Sep 2021

PONE-D-21-27242Evaluation of GENECUBE® HQ SARS-CoV-2 for anterior nasal samples and saliva samples with a new rapid examination protocolPLOS ONE

Dear Dr. Suzuki,

Thank you for submitting your manuscript to PLOS ONE. After careful consideration, we feel that it has merit but does not fully meet PLOS ONE’s publication criteria as it currently stands. Therefore, we invite you to submit a revised version of the manuscript that addresses the points raised during the review process.

ACADEMIC EDITOR: As appended below, the reviewers have raised major concern/critique and suggested further justification/work to consolidate the findings. Do go through the comments and amend the MS accordingly.

We look forward to receiving your revised manuscript.

Kind regards,

A. M. Abd El-Aty

Academic Editor

PLOS ONE

Journal Requirements:

a) Did participants provide their written or verbal informed consent to participate in this study?

"This study was supported financially by TOYOBO Co., Ltd. PSS provided the MagDEA Dx SV and magLEAD Consumable Kit for the evaluation of the rapid protocol with magLEAD. TOYOBO Co., Ltd., provided support in the form of salaries to author Akio Sugiyama and Masashi Michibuchi, lecture fees to author Hiromichi Suzuki, and advisory fees to author Hiromichi Suzuki. Hiromichi Suzuki also received advisory fees from PSS. The funder did not have any additional role in the study design, data collection and analysis, decision to publish, or preparation of the manuscript."

We note that one or more of the authors is affiliated with the funding organization, indicating the funder may have had some role in the design, data collection, analysis or preparation of your manuscript for publication; in other words, the funder played an indirect role through the participation of the co-authors. If the funding organization did not play a role in the study design, data collection and analysis, decision to publish, or preparation of the manuscript and only provided financial support in the form of authors' salaries and/or research materials, please do the following:

a. Review your statements relating to the author contributions, and ensure you have specifically and accurately indicated the role(s) that these authors had in your study. These amendments should be made in the online form.

b. Confirm in your cover letter that you agree with the following statement, and we will change the online submission form on your behalf: 

“The funder provided support in the form of salaries for authors [insert relevant initials], but did not have any additional role in the study design, data collection and analysis, decision to publish, or preparation of the manuscript. The specific roles of these authors are articulated in the ‘author contributions’ section.

"PSS provided the MagDEA Dx SV and magLEAD Consumable Kit for the evaluation of the rapid protocol with magLEAD. TOYOBO Co., Ltd., provided support in the form of salaries to author Akio Sugiyama and Masashi Michibuchi, lecture fees to author Hiromichi Suzuki, and advisory fees to author Hiromichi Suzuki. Hiromichi Suzuki also received advisory fees from PSS. The funder did not have any additional role in the study design, data collection and analysis, decision to publish, or preparation of the manuscript."

"This study was supported financially by TOYOBO Co., Ltd. PSS provided the MagDEA Dx SV and magLEAD Consumable Kit for the evaluation of the rapid protocol with magLEAD. TOYOBO Co., Ltd., provided support in the form of salaries to author Akio Sugiyama and Masashi Michibuchi, lecture fees to author Hiromichi Suzuki, and advisory fees to author Hiromichi Suzuki. Hiromichi Suzuki also received advisory fees from PSS. The funder did not have any additional role in the study design, data collection and analysis, decision to publish, or preparation of the manuscript."

7. We note that Figure 1 in your submission contain [map/satellite] images which may be copyrighted. All PLOS content is published under the Creative Commons Attribution License (CC BY 4.0), which means that the manuscript, images, and Supporting Information files will be freely available online, and any third party is permitted to access, download, copy, distribute, and use these materials in any way, even commercially, with proper attribution. For these reasons, we cannot publish previously copyrighted maps or satellite images created using proprietary data, such as Google software (Google Maps, Street View, and Earth). For more information, see our copyright guidelines: http://journals.plos.org/plosone/s/licenses-and-copyright.

Reviewers' comments:

Reviewer's Responses to Questions

**Comments to the Author**

1. Is the manuscript technically sound, and do the data support the conclusions?

Reviewer #1: Partly

Reviewer #2: Partly

Reviewer #3: Yes

2. Has the statistical analysis been performed appropriately and rigorously? 

Reviewer #1: No

Reviewer #2: Yes

Reviewer #3: Yes

3. Have the authors made all data underlying the findings in their manuscript fully available?

Reviewer #1: No

Reviewer #2: Yes

Reviewer #3: Yes

4. Is the manuscript presented in an intelligible fashion and written in standard English?

Reviewer #1: Yes

Reviewer #2: Yes

Reviewer #3: Yes

5. Review Comments to the Author

Reviewer #1: As a whole this is a good and convincing paper, but I have some concerns – both minor and major.

Journal policy concern: Authors state that there will be some restrictions on data sharing, without specifying what those are. They later state that all relevant data are in the manuscript. This seems internally contradictory.

Minor concern: Line 69: Authors state that “This reagent was approved in October, 2021.” Who was the approving body?

Major concern: Authors note that if discrepancies were found between GeneCube analysis and the RTPCR reference method, a third method was employed to resolve the discrepancy. This procedure, known as “discrepant analysis” is biased. [1–7]. Bias results from the fact that testing of non-discrepant samples (based on two assays) may, in principle, result in a discrepancy when the “resolver” test is applied; however, there is no chance to detect this discrepancy because the resolver is never applied to the non-discrepant tests. This may give the appearance that both of the initial test methods are better than they actually are. What the authors did was not as bad as it could have been since they apparently did not use the discrepant analysis to create or adjust sensitivity numbers. However, use of a composite reference standard for assessing the performance of methods does not suffer from this bias [8,9], and gives a more accurate portrayal of the relative sensitivity of both methods. While the impact of this inappropriate technique on the conclusions is negligible, its appearance in the literature promotes its further use. I think the paper would be significantly improved (and shortened somewhat) if none of the discrepant analysis is included.

Minor concern: Sampling from a single nares is to be avoided, as there is sufficient data on respiratory infections to show that sampling from both nares improves sensitivity. I include one reference, but there are many others I haven’t taken the time to look up [10]. I think the authors should note in their discussion the fact that sampling both nares is preferable.

Minor concern: The number of samples assessed in the LOD study is too small to draw strong conclusions about the LOD (see CLSI document EP7 and https://www.ncbi.nlm.nih.gov/pmc/articles/PMC2556583/). Pooling of sample types (UTM, nasal swab, saliva) to achieve N is not appropriate, since these samples may have differing sources of assay interference. I think it is fair to say the authors have approximated the LOD, however.

1. Miller WC. Bias in discrepant analysis: When two wrongs don’t make a right. Journal of Clinical Epidemiology. 1998;51: 219–231. doi:10.1016/S0895-4356(97)00264-3

2. Hadgu A. Discrepant analysis: A biased and an unscientific method for estimating test sensitivity and specificity. Journal of Clinical Epidemiology. 1999;52: 1231–1237. doi:10.1016/S0895-4356(99)00101-8

3. Hadgu A. The discrepancy in discrepant analysis. Lancet. 1996;348: 592–593. doi:10.1016/S0140-6736(96)05122-7

4. McAdam AJ. Discrepant analysis and bias: A micro-comic strip. Journal of Clinical Microbiology. American Society for Microbiology; 2017. pp. 2878–2879. doi:10.1128/JCM.00969-17

5. Hadgu A, McAdam AJ. Discrepant analysis is an inappropriate and unscientific method (multiple letters) [4]. Journal of Clinical Microbiology. 2000. pp. 4301–4302.

6. Miller WC. Can we do better than discrepant analysis for new diagnostic test evaluation? Clinical infectious diseases : an official publication of the Infectious Diseases Society of America. 1998. pp. 1186–1193. doi:10.1086/514996

7. Green TA, Black CM, Johnson RE. Evaluation of bias in diagnostic-test sensitivity and specificity estimates computed by discrepant analysis. Journal of Clinical Microbiology. 1998;36: 375–381. doi:10.1128/jcm.36.2.375-381.1998

8. Baughman AL, Bisgard KM, Cortese MM, Thompson WW, Sanden GN, Strebel PM. Utility of composite reference standards and latent class analysis in evaluating the clinical accuracy of diagnostic tests for pertussis. Clinical and Vaccine Immunology. 2008;15: 106–114. doi:10.1128/CVI.00223-07

9. Tang S, Hemyari P, Canchola JA, Duncan J. Dual composite reference standards (dCRS) in molecular diagnostic research: A new approach to reduce bias in the presence of imperfect reference. Journal of Biopharmaceutical Statistics. 2018;28: 951–965. doi:10.1080/10543406.2018.1428613

10. van Wesenbeeck L, Meeuws H, D’Haese D, Ispas G, Houspie L, van Ranst M, et al. Sampling variability between two mid-turbinate swabs of the same patient has implications for influenza viral load monitoring. Virology Journal. 2014;11. doi:10.1186/s12985-014-0233-9

Reviewer #2: The manuscript submitted by Naito et al entitled “Evaluation of GENECUBE HQ SARS-CoV-2 for anterior nasal samples and saliva samples with a new rapid examination protocol” details a study that evaluates a molecular assay for detection of SARS-CoV-2. The assay was previously evaluated for NP specimens, but this assay evaluated the use of both NS, saliva, and saliva with a rapid processing protocol. In total 359 NS were evaluated and 240 saliva samples. Overall, the results demonstrated that the assay was accurate for all specimen types tested; However, there are some discrepancies in numbers that need to be clarified to ensure the proper comparisons were made. There are also a few additional clarifications needed prior to publication listed below:

Major Comments

For the enriched positive patients that had confirmed COVID-19 results, how was this detected. Did patients come back in after the results came through or was it a rapid test. Also, by cherry picking these patients you mess with the pre-test probability and would likely improve the assay performance, especially as the LoD appears to be better for GENECUBE vs RT-PCR assay. I would suggest breaking out the results as Total, prospective, call back subjects to be clear and see effects on true prospective testing.

The definition of the rapid method is lacking in specifics. What steps were modified in the maglead extraction process that reduced the method by 20 minutes. As this is written it would not be possible to replicate the study or adopt the new method for a clinical lab to validate and use for faster TAT.

Ln223-227: This is a bit unclear. There are 3 FP results based on rapid. 1 was the FP found in 4a, but the other 2 are new. Why were these 2 specimens tested 8 times on the lightmix test? One was picked up 50% of the time and the other was 12.5%.

Ln245-250: Why is table 4a and 4b not using the NP PCR result as the reference method. This analysis was done is ST2, which should be the results used for the evaluation and not comparing saliva tested on the two platforms (Is the reference PCR validated for saliva?). If you use the presentation of data in table 4a and 4b this would suggest that saliva with GENECUBE is more sensitive, but if you compare it to the PCR from NP (gold standard) then it seems like it would be slightly less sensitive of a sample type. I would delete 4a and 4b and replace with data comparing to NP. This is the same for NS comparison data as ln 257 indicates that PPA was actually 88.1% compared to the 98.1% you present in table 3.

Minor Comments

Ln 73: Was the pos and negative concordance both 99.7% in the NP study as you only give 2 values for the overall, pos-, neg concordance.

Were discordant specimens tested on the Xpert run as package insert (i.e. testing directly from VTM).

Table 2: The current layout is a bit confusing at first with the spacing for “Standard method with magLead…”

LoD study: The standard method is to perform 20 replicates at each concentration in pooled negatives and the LoD is then defined as the lowest concentration that was 19/20. What was the rationale for using multiple matrices and at the end having 24 tests for GENECUBE and 12 for the reference method.

Ln197: Was the copies/mL determined based on the CT value and the LoD of the specimen? If so what was the CT value for the reference test and the Xpert assay.

Why were the saliva samples frozen initially? Could this have skewed results as there is some data that a freeze thaw can help sensitivity of saliva samples.

Reviewer #3: Thank you for inviting me to peer review this manuscript. The authors have studied the GENECUBE ® HQ SARS-CoV-2 using anterior nasal samples and saliva samples by using a new protocol. This study aims to evaluate the detection of SARS-CoV-2 with saliva samples for the first time and using a rapid protocol. Here are some comments which could be considers to improve the manuscript:

1. Line 69 “This reagent was approved in October, 2021.”. The date needs to be corrected.

2. Line 86, the situation of the infected cases is not explained. How did the authors select them and in which phase of disease they were? Did they have symptoms or they were asymptomatic cases?

3. It seems that the authors have collected the anterior nasal samples and saliva samples from different cases. It was better to take both sample types from each studied case to be able to compare them and investigate the accuracy of the samples to detect the infection. (It seems that such work is mentioned very briefly in the Discussion)

4. There is no explanation for positive and negative controls.

5. New Method and Standard Method are not explained in the Materials and Methods. In Line 133, “For the rapid protocol, in the preparation of saliva samples, purification and extraction processes were adjusted, and the total process time was shortened to approximately 10 minutes.” How this happened and what is the main difference making the novel method this short?

6. The obtained resulted are not discussed deeply. As an example, Line 181 “The detection rate was 100% down to 1000 copies/mL for pooled …” by considering these results, is this method useful for early detection or it is just useful for the chronic cases? How could a Dr. decide this strategy is a good choice for a specific case (like asymptomatic cases, accurate or chronic infections)?

7. Line 186. Compared with the standard test, how could the Rapid Strategy decrease the LoD of the nasopharyngeal samples from 1000 copies/mL to 500 copies/mL? (not discussed again).

8. Line 245 “In our current study, we used 60 saliva samples with positive nasopharyngeal sample results for SARS-CoV-2 (Supplementary Table 2), and the rapid protocol detected SARS-CoV-2 in most samples (98.3%; 59/60)”. It is the first time that the authors are refer to performing some tests on the nasopharyngeal and saliva samples taken from one case. It is better to be explained (materials and methods) and reported (results) in the previous sections before the Discussion.

9. The authors could also add graph(s) to report the results in a more useful way.

6. PLOS authors have the option to publish the peer review history of their article (what does this mean?). If published, this will include your full peer review and any attached files.

Reviewer #1: No

Reviewer #2: No

Reviewer #3: **Yes: **Tina Shaffaf

---

## [Author Response · Author response to Decision Letter 0]

19 Oct 2021

Suggestion: Please ensure that your manuscript meets PLOS ONE's style requirements, including those for file naming. The PLOS ONE style templates can be found at 

Response: We revised our manuscript to meet PLOS ONE's style requirements; changes to the text are shown with red marking.

Suggestion: Please amend your current ethics statement to address the following concerns:

a) Did participants provide their written or verbal informed consent to participate in this study?

Response: Informed consent was verbally obtained from patients for their participation in the respective parts of the current research, and participant consent was documented in the electronic chart of each participant. Written informed consent was not obtained in order to avoid infection transmission through the consent forms. The ethics committee approved the present study with the current protocol, including the method of obtaining informed consent. We have now mentioned this in the Methods section.

Suggestion: We note that the grant information you provided in the ‘Funding Information’ and ‘Financial Disclosure’ sections do not match. When you resubmit, please ensure that you provide the correct grant numbers for the awards you received for your study in the ‘Funding Information’ section.

Response: We revised both the ‘Funding Information’ and ‘Financial Disclosure’ sections to be consistent. There is no grant number in current study. 

Suggestion: 

Thank you for stating the following financial disclosure: 

"This study was supported financially by TOYOBO Co., Ltd. PSS provided the MagDEA Dx SV and magLEAD Consumable Kit for the evaluation of the rapid protocol with magLEAD. TOYOBO Co., Ltd., provided support in the form of salaries to author Akio Sugiyama and Masashi Michibuchi, lecture fees to author Hiromichi Suzuki, and advisory fees to author Hiromichi Suzuki. Hiromichi Suzuki also received advisory fees from PSS. The funder did not have any additional role in the study design, data collection and analysis, decision to publish, or preparation of the manuscript."

We note that one or more of the authors is affiliated with the funding organization, indicating the funder may have had some role in the design, data collection, analysis or preparation of your manuscript for publication: in other words, the funder played an indirect role through the participation of the co-authors. If the funding organization did not play a role in the study design, data collection and analysis, decision to publish, or preparation of the manuscript and only provided financial support in the form of authors' salaries and/or research materials, please do the following:

a. Review your statements relating to the author contributions, and ensure you have specifically and accurately indicated the role(s) that these authors had in your study. These amendments should be made in the online form.

b. Confirm in your cover letter that you agree with the following statement, and we will change the online submission form on your behalf: 

Response: We have now reviewed our statements relating to the authors’ contributions and ensured that we have specifically and accurately indicated the roles that those authors played in our study. In addition, we confirmed in our cover letter that we agree with the statement below. Please change the online submission form on our behalf.

“The funder provided support in the form of salaries for authors A. S. and M. M. but did not play any additional role in the study design, data collection or analysis, decision to publish, or preparation of the manuscript. The specific roles of these authors are articulated in the ‘author contributions’ section.”

Suggestion: Thank you for stating the following in the Acknowledgments Section of your manuscript: 

"PSS provided the MagDEA Dx SV and magLEAD Consumable Kit for the evaluation of the rapid protocol with magLEAD. TOYOBO Co., Ltd., provided support in the form of salaries to author Akio Sugiyama and Masashi Michibuchi, lecture fees to author Hiromichi Suzuki, and advisory fees to author Hiromichi Suzuki. Hiromichi Suzuki also received advisory fees from PSS. The funder did not have any additional role in the study design, data collection and analysis, decision to publish, or preparation of the manuscript."

"This study was supported financially by TOYOBO Co., Ltd. PSS provided the MagDEA Dx SV and magLEAD Consumable Kit for the evaluation of the rapid protocol with magLEAD. TOYOBO Co., Ltd., provided support in the form of salaries to author Akio Sugiyama and Masashi Michibuchi, lecture fees to author Hiromichi Suzuki, and advisory fees to author Hiromichi Suzuki. Hiromichi Suzuki also received advisory fees from PSS. The funder did not have any additional role in the study design, data collection and analysis, decision to publish, or preparation of the manuscript."

Please include your amended statements within your cover letter: we will change the online submission form on your behalf.

Response: We removed any funding-related text from our manuscript and included our amended statements within our cover letter. We have provided our amended Funding Statement below. Please change the online submission form on our behalf.

"This study was supported financially by TOYOBO Co., Ltd. PSS provided the MagDEA Dx SV and magLEAD Consumable Kit for the evaluation of the rapid protocol with magLEAD. TOYOBO Co., Ltd., provided support in the form of salaries for authors Akio Sugiyama and Masashi Michibuchi, lecture fees for author Hiromichi Suzuki, and advisory fees for author Hiromichi Suzuki. Hiromichi Suzuki also received advisory fees from PSS. The funder did not play any additional role in the study design, data collection or analysis, decision to publish, or preparation of the manuscript."

Suggestion: 

In your Data Availability statement, you have not specified where the minimal data set underlying the results described in your manuscript can be found. PLOS defines a study's minimal data set as the underlying data used to reach the conclusions drawn in the manuscript and any additional data required to replicate the reported study findings in their entirety. All PLOS journals require that the minimal data set be made fully available. For more information about our data policy, please see http://journals.plos.org/plosone/s/data-availability. Upon re-submitting your revised manuscript, please upload your study’s minimal underlying data set as either Supporting Information files or to a stable, public repository and include the relevant URLs, DOIs, or accession numbers within your revised cover letter. For a list of acceptable repositories, please see http://journals.plos.org/plosone/s/data-availability#loc-recommended-repositories. Any potentially identifying patient information must be fully anonymized. Important: If there are ethical or legal restrictions to sharing your data publicly, please explain these restrictions in detail. Please see our guidelines for more information on what we consider unacceptable restrictions to publicly sharing data: http://journals.plos.org/plosone/s/data-availability#loc-unacceptable-data-access-restrictions. Note that it is not acceptable for the authors to be the sole named individuals responsible for ensuring data access.

Response: We have now uploaded our study’s minimal underlying data set as S1 Table and S2 Table in the supporting information. Please update our Data Availability statement to reflect the information we provided in our cover letter.

Suggestion: 

We note that Figure 1 in your submission contain [map/satellite] images which may be copyrighted. All PLOS content is published under the Creative Commons Attribution License (CC BY 4.0), which means that the manuscript, images, and Supporting Information files will be freely available online, and any third party is permitted to access, download, copy, distribute, and use these materials in any way, even commercially, with proper attribution. For these reasons, we cannot publish previously copyrighted maps or satellite images created using proprietary data, such as Google software (Google Maps, Street View, and Earth). For more information, see our copyright guidelines: http://journals.plos.org/plosone/s/licenses-and-copyright.

Response: We have now uploaded the completed content permission form as an “Other” file along with the submission. In addition, we added the following text to the caption of the copyrighted figure:

“For magLEAD, the picture was reprinted from [13] under a CC BY license with the permission of Precision System Science Co., Ltd., 2016” and “For GENECUBE, てthe picture was reprinted from [14] under a CC BY license with the permission of TOYOBO Co., Ltd., 2021”.

[13] Precision System Science Co., Ltd. magLEAD 6gC & magLEAD 12gC High-Quality, Low Cost, Automated Nucleic Acid Extraction System. 2021 [Cited 5 August 2021]. Available from https://www.pss.co.jp/product/magtration/lead6-12gc.html

[14] TOYOBO Co., Ltd. Fully Automated Gene Analyzer GENECUBE® (model C). 2021 [Cited 5 August 2021]. Available from https://www.toyobo.co.jp/products/bio/gene/genecube_c/index.html.

5. Review Comments to the Author

Reviewer #1: As a whole this is a good and convincing paper, but I have some concerns – both minor and major.　

Response: We appreciate your review and suggestions for our manuscript. We have now revised the manuscript based on the suggestions, with changes shown in red.

Suggestion: Journal policy concern: Authors state that there will be some restrictions on data sharing, without specifying what those are. They later state that all relevant data are in the manuscript. This seems internally contradictory.

Response: We have now added all of our data as supplementary tables 1 and 2 and modified our statement. 

Suggestion: Minor concern: Line 69: Authors state that “This reagent was approved in October, 2021.” Who was the approving body?

Response: The reagent was approved by the Ministry of Health, Labour and Welfare in Japan in October 2020. We changed the year of approval from 2021 to 2020, which was an error in our description. 

Major concern: Authors note that if discrepancies were found between GeneCube analysis and the RTPCR reference method, a third method was employed to resolve the discrepancy. This procedure, known as “discrepant analysis” is biased. [1–7]. Bias results from the fact that testing of non-discrepant samples (based on two assays) may, in principle, result in a discrepancy when the “resolver” test is applied: however, there is no chance to detect this discrepancy because the resolver is never applied to the non-discrepant tests. This may give the appearance that both of the initial test methods are better than they actually are. What the authors did was not as bad as it could have been since they apparently did not use the discrepant analysis to create or adjust sensitivity numbers. However, use of a composite reference standard for assessing the performance of methods does not suffer from this bias [8,9], and gives a more accurate portrayal of the relative sensitivity of both methods. While the impact of this inappropriate technique on the conclusions is negligible, its appearance in the literature promotes its further use. I think the paper would be significantly improved (and shortened somewhat) if none of the discrepant analysis is included.

Response: We completely agree with your comment and admit that this is a limitation of our current study: Reference real-time RT-PCR was used for the comparison, and a discrepant analysis was used for the validation, which can cause bias unless a composite reference standard is used as a reference. While the current study evaluated the concordance with the reference real-time RT-PCR findings, the sensitivity and specificity of the GENECUBE® examination were not accurately confirmed. Regarding the discrepant analysis, the description was deleted from the Results section of the manuscript, and the table concerning the results of the discrepancy analysis of saliva was moved to the supplementary materials. 

Minor concern: Sampling from a single nares is to be avoided, as there is sufficient data on respiratory infections to show that sampling from both nares improves sensitivity. I include one reference, but there are many others I haven’t taken the time to look up [10]. I think the authors should note in their discussion the fact that sampling both nares is preferable.

Response: We have now mentioned this in the Discussion section as follows: “[…] however, the viral loads obtained with nasal sampling differ between nares, so sampling from both nasal cavities is preferable.” 

Minor concern: The number of samples assessed in the LOD study is too small to draw strong conclusions about the LOD (see CLSI document EP7 and https://www.ncbi.nlm.nih.gov/pmc/articles/PMC2556583/). Pooling of sample types (UTM, nasal swab, saliva) to achieve N is not appropriate, since these samples may have differing sources of assay interference. I think it is fair to say the authors have approximated the LOD, however.

Response: The current evaluation was insufficient to determine the LODs. We have now revised the text concerning the evaluation and withdrew our strong conclusion concerning LODs, which mentioned the superiority of the GENECUBE examination. 

1. Miller WC. Bias in discrepant analysis: When two wrongs don’t make a right. Journal of Clinical Epidemiology. 1998:51: 219–231. doi:10.1016/S0895-4356(97)00264-3

2. Hadgu A. Discrepant analysis: A biased and an unscientific method for estimating test sensitivity and specificity. Journal of Clinical Epidemiology. 1999:52: 1231–1237. doi:10.1016/S0895-4356(99)00101-8

3. Hadgu A. The discrepancy in discrepant analysis. Lancet. 1996:348: 592–593. doi:10.1016/S0140-6736(96)05122-7

4. McAdam AJ. Discrepant analysis and bias: A micro-comic strip. Journal of Clinical Microbiology. American Society for Microbiology: 2017. pp. 2878–2879. doi:10.1128/JCM.00969-17

5. Hadgu A, McAdam AJ. Discrepant analysis is an inappropriate and unscientific method (multiple letters) [4]. Journal of Clinical Microbiology. 2000. pp. 4301–4302.

6. Miller WC. Can we do better than discrepant analysis for new diagnostic test evaluation? Clinical infectious diseases : an official publication of the Infectious Diseases Society of America. 1998. pp. 1186–1193. doi:10.1086/514996

7. Green TA, Black CM, Johnson RE. Evaluation of bias in diagnostic-test sensitivity and specificity estimates computed by discrepant analysis. Journal of Clinical Microbiology. 1998:36: 375–381. doi:10.1128/jcm.36.2.375-381.1998

8. Baughman AL, Bisgard KM, Cortese MM, Thompson WW, Sanden GN, Strebel PM. Utility of composite reference standards and latent class analysis in evaluating the clinical accuracy of diagnostic tests for pertussis. Clinical and Vaccine Immunology. 2008:15: 106–114. doi:10.1128/CVI.00223-07

9. Tang S, Hemyari P, Canchola JA, Duncan J. Dual composite reference standards (dCRS) in molecular diagnostic research: A new approach to reduce bias in the presence of imperfect reference. Journal of Biopharmaceutical Statistics. 2018:28: 951–965. doi:10.1080/10543406.2018.1428613

10. van Wesenbeeck L, Meeuws H, D’Haese D, Ispas G, Houspie L, van Ranst M, et al. Sampling variability between two mid-turbinate swabs of the same patient has implications for influenza viral load monitoring. Virology Journal. 2014:11. doi:10.1186/s12985-014-0233-9

Reviewer #2: The manuscript submitted by Naito et al entitled “Evaluation of GENECUBE HQ SARS-CoV-2 for anterior nasal samples and saliva samples with a new rapid examination protocol” details a study that evaluates a molecular assay for detection of SARS-CoV-2. The assay was previously evaluated for NP specimens, but this assay evaluated the use of both NS, saliva, and saliva with a rapid processing protocol. In total 359 NS were evaluated and 240 saliva samples. Overall, the results demonstrated that the assay was accurate for all specimen types tested: However, there are some discrepancies in numbers that need to be clarified to ensure the proper comparisons were made. There are also a few additional clarifications needed prior to publication listed below:

Response: We appreciate your review and suggestions for our manuscript. We have now revised the manuscript based on the suggestions, with changes shown in red.

Major Comments

Suggestion: For the enriched positive patients that had confirmed COVID-19 results, how was this detected. Did patients come back in after the results came through or was it a rapid test. Also, by cherry picking these patients you mess with the pre-test probability and would likely improve the assay performance, especially as the LoD appears to be better for GENECUBE vs RT-PCR assay. I would suggest breaking out the results as Total, prospective, call back subjects to be clear and see effects on true prospective testing.

Response: We have now included additional data in Table 4a, which excluded anterior nasal samples obtained from confirmed COVID-19 patients, in the Results section. 

Suggestion: The definition of the rapid method is lacking in specifics. What steps were modified in the magLEAD extraction process that reduced the method by 20 minutes. As this is written it would not be possible to replicate the study or adopt the new method for a clinical lab to validate and use for faster TAT.

Response: We added Table 1 to show the results of the comparison of the standard and rapid protocols for each magLEAD purification processes in the Methods section. The new extraction method is now commercially available in Japan as MagDEA Dx SV 200 for GENECUBE® 12gC, so researchers can validate the current findings. We have now mentioned this in the revised manuscript.

Suggestion: Ln223-227: This is a bit unclear. There are 3 FP results based on rapid. 1 was the FP found in 4a, but the other 2 are new. Why were these 2 specimens tested 8 times on the lightmix test? One was picked up 50% of the time and the other was 12.5%.

Response: All three discordant saliva samples were obtained from COVID-19 patients confirmed with simultaneously obtained nasopharyngeal samples (supplementary table 2). For #11, #39, additional RT-PCR (E-gene) was performed with two additional purification methods, as RT-PCR (E-gene) was negative for the initial purified sample. Multiple tests (eight times) were performed because the viral loads of SARS-CoV-2 in the saliva samples were considered to be very low. These data are considered to be supplementary, so we moved the discrepancy analysis to a supplementary table. 

Suggestion: Ln245-250: Why is table 4a and 4b not using the NP PCR result as the reference method. This analysis was done is ST2, which should be the results used for the evaluation and not comparing saliva tested on the two platforms (Is the reference PCR validated for saliva?). If you use the presentation of data in table 4a and 4b this would suggest that saliva with GENECUBE is more sensitive, but if you compare it to the PCR from NP (gold standard) then it seems like it would be slightly less sensitive of a sample type. I would delete 4a and 4b and replace with data comparing to NP. This is the same for NS comparison data as ln 257 indicates that PPA was actually 88.1% compared to the 98.1% you present in table 3.

Response: The reference method has been validated for saliva, and we added a reference to the Method section. For anterior nasal samples, all samples were simultaneously obtained with nasopharyngeal samples, and we added a comparison between GENECUBE® HQ SARS-CoV-2 with anterior nasal samples and real-time RT-PCR with nasopharyngeal samples to the Results section. For saliva samples, while positive saliva samples were simultaneously obtained with nasopharyngeal samples, negative saliva samples were not simultaneously obtained with nasopharyngeal samples, so we cannot make the table suggested. We added the limitation in discussion section.

Minor Comments

Suggestion: Ln 73: Was the pos and negative concordance both 99.7% in the NP study as you only give 2 values for the overall, pos-, neg concordance.

Response: This was our mistake. We have now added the data concerning total concordance in red. 

Suggestion: Were discordant specimens tested on the Xpert run as package insert (i.e. testing directly from VTM).

Response: We tested the discordant specimens directly from UTM according to the manufacturer’s instructions included on the package insert. We have now added the description to the Methods section in red. 

Suggestion: Table 2: The current layout is a bit confusing at first with the spacing for “Standard method with magLead…”

Response: We changed the layout to improve the appearance. 

Suggestion: LoD study: The standard method is to perform 20 replicates at each concentration in pooled negatives and the LoD is then defined as the lowest concentration that was 19/20. What was the rationale for using multiple matrices and at the end having 24 tests for GENECUBE and 12 for the reference method.　

Response: As proposed, it was inappropriate to draw conclusions about LoDs with the current LoD study, although the current protocol is useful for investigating the influence of the respiratory sample matrix on the analytical performance of GENECUBE examinations and is also useful for estimating LoDs. We have now revised the expression to “estimated LoDs”, and the assessment of the results was added to the Discussion section in red. 

Suggestion: Ln197: Was the copies/mL determined based on the CT value and the LoD of the specimen? If so what was the CT value for the reference test and the Xpert assay.

Response: For quantitative RT-PCR, the Ct value of reference RT-PCR with 5, 50, and 500 copies/reaction of positive control was used. We have now added the quantitative method to the Methods section in red. 

Suggestion: Why were the saliva samples frozen initially? Could this have skewed results as there is some data that a freeze thaw can help sensitivity of saliva samples.

Response: The current study was performed with two GENECUBE examinations (standard protocol and rapid protocol) and reference RT-PCR. Due to a lack of human resources, we were unable to perform GENECUBE examinations and reference RT-PCR with fresh saliva samples. We have now mentioned this as a limitation in the Discussion section. While the analytical performance for the detection of SARS-CoV-2 has been reported to be equal between fresh and frozen saliva samples [11], the freeze-thaw method has been known as one of the RNA/DNA extraction methods [12] which might influence the sensitivity [13]. We have now mentioned this in the Discussion section. 

11. Fukumoto T, Iwasaki S, Fujisawa S, Hayasaka K, Sato K, Oguri S, et al. Efficacy of a novel SARS-CoV-2 detection kit without RNA extraction and purification. Int J Infect Dis. 2020:98:16-17: https://doi.org/10.1016/j.ijid.2020.06.074

12. Paz S, Mauer C, Ritchie A, Robishaw JD, Caputi M. A simplified SARS-CoV-2 detection protocol for research laboratories. PLoS One. 2020:15: e0244271: https://doi.org/10.1371/journal.pone.0244271

13. Ott IM, Strine MS, Watkins AE, Boot M, Kalinich CC, Harden CA, et al. Simply saliva: stability of SARS-CoV-2 detection negates the need for expensive collection devices. madRxiv. 2020: https://doi.org/10.1101/2020.08.03.20165233

Reviewer #3: Thank you for inviting me to peer review this manuscript. The authors have studied the GENECUBE ® HQ SARS-CoV-2 using anterior nasal samples and saliva samples by using a new protocol. This study aims to evaluate the detection of SARS-CoV-2 with saliva samples for the first time and using a rapid protocol. Here are some comments which could be considers to improve the manuscript:

Response: We appreciate your review and suggestions for our manuscript. We have now revised the manuscript based on the suggestions, with changes shown in red.

Suggestion: Line 69 “This reagent was approved in October, 2021.”. The date needs to be corrected.

Response: This was our mistake. We have now changed the year of approval from 2021 to 2020.

Suggestion: Line 86, the situation of the infected cases is not explained. How did the authors select them and in which phase of disease they were? Did they have symptoms or they were asymptomatic cases?

Response: We have now clarified the situation of the patients in the Methods section and the number of patients with symptoms in the Results section. 

Suggestion: It seems that the authors have collected the anterior nasal samples and saliva samples from different cases. It was better to take both sample types from each studied case to be able to compare them and investigate the accuracy of the samples to detect the infection. (It seems that such work is mentioned very briefly in the Discussion)

Response: As you suggested, saliva samples were not simultaneously collected with anterior nasal samples in this study; we therefore cannot compare the sensitivity. We have now mentioned this as a limitation in the Discussion section. 

Suggestion: There is no explanation for positive and negative controls.

Response: EDX SARS-CoV-2 Standard (Bio-Rad Laboratories, Inc., Hercules, CA, USA) and sterile purified water (Merck & Co., Inc., Kenilworth, NJ, USA) were used as positive and negative controls, respectively. We have now mentioned this in the Methods section. 

Suggestion: New Method and Standard Method are not explained in the Materials and Methods. In Line 133, “For the rapid protocol, in the preparation of saliva samples, purification and extraction processes were adjusted, and the total process time was shortened to approximately 10 minutes.” How this happened and what is the main difference making the novel method this short?

Response: We added Table 1 to show the results of the comparison of the standard and rapid protocols for each magLEAD purification processes in the Methods section. 

Suggestion: The obtained resulted are not discussed deeply. As an example, Line 181 “The detection rate was 100% down to 1000 copies/mL for pooled …” by considering these results, is this method useful for early detection or it is just useful for the chronic cases? How could a Dr. decide this strategy is a good choice for a specific case (like asymptomatic cases, accurate or chronic infections)?

Response: We have now described our assessment of the rapid method and the current limitations associated with the study of anterior nasal samples and saliva samples in the Discussion section. The rapid protocol with a GENECUBE examination for saliva is considered useful for the rapid accurate identification of SARS-CoV-2-infected patients when collecting nasopharyngeal samples is difficult. 

Suggestion: Line 186. Compared with the standard test, how could the Rapid Strategy decrease the LoD of the nasopharyngeal samples from 1000 copies/mL to 500 copies/mL? (not discussed again).

Response: The total detection rate in the estimated LOD study was considered equal between the two protocols, although the detection rate was slightly higher than that with the standard method in a comparative study of saliva samples. This slight difference might be due to differences in the centrifugation condition (Fig. 1) and number of washes performed (Table 1), which can result in the loss of virus and RNA; however, the difference was not proven in the estimated LOD study, and the superiority with respect to the analytical performance cannot be concluded based on the present findings. We have now mentioned this in the Discussion. 

Suggestion: Line 245 “In our current study, we used 60 saliva samples with positive nasopharyngeal sample results for SARS-CoV-2 (Supplementary Table 2), and the rapid protocol detected SARS-CoV-2 in most samples (98.3%: 59/60)”. It is the first time that the authors are refer to performing some tests on the nasopharyngeal and saliva samples taken from one case. It is better to be explained (materials and methods) and reported (results) in the previous sections before the Discussion.

Response: We have now described the simultaneous sampling collection in the Methods section and the number in the Results section. 

Suggestion: The authors could also add graph(s) to report the results in a more useful way.

Response: We have now added a supplementary figure (S1) to support the viewing of table 2 in the supporting information file.

---

## [Decision Letter · Decision Letter 1]

29 Nov 2021

PONE-D-21-27242R1The evaluation of the utility of the GENECUBE HQ SARS-CoV-2 for anterior nasal samples and saliva samples with a new rapid examination protocolPLOS ONE

Dear Dr. Suzuki,

Thank you for submitting your manuscript to PLOS ONE. After careful consideration, we feel that it has merit but does not fully meet PLOS ONE’s publication criteria as it currently stands. Therefore, we invite you to submit a revised version of the manuscript that addresses the points raised during the review process.

We look forward to receiving your revised manuscript.

Kind regards,

A. M. Abd El-Aty

Academic Editor

PLOS ONE

Journal Requirements:

Reviewers' comments:

Reviewer's Responses to Questions

**Comments to the Author**

1. If the authors have adequately addressed your comments raised in a previous round of review and you feel that this manuscript is now acceptable for publication, you may indicate that here to bypass the “Comments to the Author” section, enter your conflict of interest statement in the “Confidential to Editor” section, and submit your "Accept" recommendation.

Reviewer #1: All comments have been addressed

Reviewer #2: All comments have been addressed

Reviewer #3: All comments have been addressed

2. Is the manuscript technically sound, and do the data support the conclusions?

Reviewer #1: (No Response)

Reviewer #2: Yes

Reviewer #3: Yes

3. Has the statistical analysis been performed appropriately and rigorously? 

Reviewer #1: (No Response)

Reviewer #2: Yes

Reviewer #3: I Don't Know

4. Have the authors made all data underlying the findings in their manuscript fully available?

Reviewer #1: (No Response)

Reviewer #2: Yes

Reviewer #3: Yes

5. Is the manuscript presented in an intelligible fashion and written in standard English?

Reviewer #1: (No Response)

Reviewer #2: Yes

Reviewer #3: Yes

6. Review Comments to the Author

Reviewer #1: (No Response)

Reviewer #2: Thank you for the changes to the manuscript, which improves the clarity of study. One minor suggestion I would have is to add a limitation about the lack of low viral load samples tested in the evaluation. In ln 202 only 3 samples were CT >30. This is a small proportion of your positives and that could have improved concordance as too few were near the limit of detection.

Reviewer #3: (No Response)

7. PLOS authors have the option to publish the peer review history of their article (what does this mean?). If published, this will include your full peer review and any attached files.

Reviewer #1: No

Reviewer #2: No

Reviewer #3: No

---

## [Author Response · Author response to Decision Letter 1]

7 Dec 2021

Suggestion: Thank you for the changes to the manuscript, which improves the clarity of study. One minor suggestion I would have is to add a limitation about the lack of low viral load samples tested in the evaluation. line 202, only 3 samples were CT >30. This is a small proportion of your positives and that could have improved concordance as too few were near the limit of detection.

Response: Thank you very much for your suggestion. As you pointed out, the proportion of low viral load samples (Ct ≥ 30) was small for the evaluation of the GENECUBE® examination using anterior nasal samples, which could have improved thus concordance rates of the current study. We therefore added this point as one limitation associated with our study in the limitation section of the discussion.

---

## [Decision Letter · Decision Letter 2]

17 Dec 2021

The evaluation of the utility of the GENECUBE HQ SARS-CoV-2 for anterior nasal samples and saliva samples with a new rapid examination protocol

PONE-D-21-27242R2

Dear Dr. Suzuki,

We’re pleased to inform you that your manuscript has been judged scientifically suitable for publication and will be formally accepted for publication once it meets all outstanding technical requirements.

Kind regards,

A. M. Abd El-Aty

Academic Editor

PLOS ONE

Additional Editor Comments (optional):

Reviewers' comments:

Reviewer's Responses to Questions

**Comments to the Author**

1. If the authors have adequately addressed your comments raised in a previous round of review and you feel that this manuscript is now acceptable for publication, you may indicate that here to bypass the “Comments to the Author” section, enter your conflict of interest statement in the “Confidential to Editor” section, and submit your "Accept" recommendation.

Reviewer #2: All comments have been addressed

2. Is the manuscript technically sound, and do the data support the conclusions?

Reviewer #2: Yes

3. Has the statistical analysis been performed appropriately and rigorously? 

Reviewer #2: Yes

4. Have the authors made all data underlying the findings in their manuscript fully available?

Reviewer #2: Yes

5. Is the manuscript presented in an intelligible fashion and written in standard English?

Reviewer #2: Yes

6. Review Comments to the Author

Reviewer #2: (No Response)

7. PLOS authors have the option to publish the peer review history of their article (what does this mean?). If published, this will include your full peer review and any attached files.

Reviewer #2: No

---

## [Editor Report · Acceptance letter]

22 Dec 2021

PONE-D-21-27242R2 

The evaluation of the utility of the GENECUBE HQ SARS-CoV-2 for anterior nasal samples and saliva samples with a new rapid examination protocol 

Dear Dr. Suzuki:

I'm pleased to inform you that your manuscript has been deemed suitable for publication in PLOS ONE. Congratulations! Your manuscript is now with our production department. 

Kind regards, 

on behalf of

Prof. A. M. Abd El-Aty 

Academic Editor

PLOS ONE